# Accounts of preventative coping: an interview study of stroke survivors on general practice registers

Peter Scott Reid, Emma Neville, Frances Cater, Ricky Mullis, Jonathan Mant, Robbie Duschinsky

Primary Care Unit, University of Cambridge, Cambridge, UK

**Correspondence to**
Dr Robbie Duschinsky;
rd522@medschl.cam.ac.uk

## ABSTRACT

**Objectives** Preventative coping is an underexplored aspect of coping behaviour. Specifically, coping is a key concern in stroke survivor accounts, but this has yet to be investigated with reference to secondary prevention.
**Design** Secondary analysis of a qualitative data set comprising semistructured interviews of 22 stroke survivors recruited from five general practices in the East of England. The topic guide included exploration of advice and support given by their doctor on medication and lifestyle. The interviews were coded using thematic analysis.
**Results** The accounts emphasised individual responsibility. Two key themes were identified, which foregrounded the role of self-concept for coping: (a) striving to be 'good', (b) appeal to ideas of 'personality'. In the former, preventative behaviour was depicted in moralistic terms, with the doctor as an adjudicator. In the latter, participants attributed their coping behaviour to their personality, which might help or hinder these efforts.
**Conclusions** We highlight that coping was characterised by survivors as something enacted by the individual self, and consider how constructions of self may impact preventative coping efforts.

## INTRODUCTION

Mortality from stroke in the UK is declining but the prevalence of stroke is rising due to improved survival rates and an ageing population.[1] Stroke frequently results in major life changes and can influence a person physically, psychologically and socially.[2] [3] Stroke survivors living in the community often report they have little support—but also anticipate little support since they feel they are obliged to take personal responsibility for managing their own recovery.[4] [5] For instance, Fletcher *et al* found themes of self-responsibility and the power of medical authority in their study of stroke survivors.[6] They interpreted their findings in terms of Foucauldian ideas of power, in which survivors were enjoined to discipline themselves in order to effect a successful recovery, and relieve pressure on health and social care services. Commentators have suggested that this construction of stroke

## STRENGTHS AND LIMITATIONS OF THIS STUDY

⇒ The study drew upon detailed qualitative characterisation of the perceptions of stroke survivors of aspects of coping with reference to secondary prevention.
⇒ The analytical approach foregrounded both lines of consistency and variation in how stroke survivors spoke about their experiences.
⇒ The data relate to participants' perceptions of their coping, rather than their coping practices.
⇒ From a single interview with each participant, we cannot draw firm conclusions about how coping is embedded in aspects of participants' lives over time.

survivorship reflects broader discourses on chronic illness that highlight individual responsibility and self-sufficiency, under guidance from distant experts but with limited service provision.[7] [8]

Understanding stroke survivors' perspectives through the lens of 'coping' can add knowledge of the needs of this patient group as well as inform services for stroke survivors. Perhaps the classic definition of coping was offered in the 1980s by Lazarus and Folkman, who characterise it as the 'thoughts and behaviours that people use to manage the internal and external demands of situations that are appraised as stressful'.[9] 'Coping' used in this way is the attempt to manage challenges in life, rather than necessarily being successful in that attempt. Most work on coping has addressed the challenge of how individuals manage current problems. Yet in recent years scholarship has increasingly emphasised coping as frequently a joint undertaking.[10] [11] Sometimes this is explicit, with actors aware of the coordinated or collective work 'we' do to cope. Sometimes this is implicit, with actors less able to willingly acknowledge the contribution of others to 'my' coping with the demands of situations. Such accounts have been frequently found to be shaped by perceptions of the source of the

stressful demands, and on attributions about others, and the extent to which the demands are felt to be a disruption or in continuity with past experiences.[12]

Much qualitative research on stroke survivors has been focused on the immediate period up to 1 year following the stroke, thus prioritising the initial stages of disruption.[13–17] A few studies have conducted later follow-up, but with a focus on how survivors deal with the effects of the initial stroke and not on secondary prevention.[18 19] More recently, secondary prevention following stroke has seen attention by a small number of studies and reviews,[20–22] but these acknowledge that the topic remains underexplored. In particular, there has been relatively little attention to 'preventative coping', which can be defined as the efforts individuals make to prepare for uncertain events in the long run by building up resources that will result in less strain in the future by minimising the event's severity.[23]

Guidelines emphasise the importance of changes to lifestyle and the role of medication in reducing the risk of having a second stroke.[24] While they stress the importance of a personalised approach, this is in terms of what to recommend rather than how to optimise chances that the recommendations will be adopted. Analysis of reasons for this suboptimal use that explore patient perspectives have tended to focus on aspects such as self-care ability, knowledge and understanding, rather than the role of preventative coping strategies. Furthermore, what attention is given to preventative coping tends to treat it as the action of individuals, rather than a joint or jointly coordinated activity with others. Better understanding of preventative coping strategies might lay the ground for more nuanced ways of developing approaches to supporting secondary prevention after stroke. It was recognised that a qualitative data set of interviews of stroke survivors on general practice stroke registers that had been performed to inform a primary care-based model of longer term care for stroke might provide insights into these issues. This article therefore sets out to answer the question: how do stroke survivors characterise their coping process in the context of secondary prevention?

## METHODS
### Study design and participants
A secondary analysis was conducted on transcripts of semistructured interviews carried out to inform the design of an intervention aimed at improving primary care after stroke. The aim of the original study was to understand how primary care meets stroke survivors' needs, and how it could better address them in the future. Participants were recruited from five general practices in the East of England. Our target was to recruit 20 stroke survivors. The participating general practices generated a list of people on their stroke register, and a random sample of these were reviewed by the local general practitioner (GP) to confirm eligibility. Inclusion criteria were: (1) confirmed diagnosis of stroke from survivor's records; (2) has a

| Table 1 | Participant information |
| --- | --- |
| Total subjects | 22 |
| Men | 11 |
| Women | 11 |
| Age | |
| Range | 49–93 |
| Mean | 73 |
| Median | 70 |
| SD | 11.8 |
| Barthel Index as a measure of disability (total score, out of 100) | |
| Range | 50–100 |
| Mean | 89.8 |
| Median | 100.0 |
| Length of time since stroke | |
| Range | 3 months to 22 years 6 months |
| Mean | 7 years |
| Median | 3 years |
| SD | 5 years 11 months |

good understanding of English; and (3) has the capacity to provide written informed consent. Exclusion criterion was survivors considered by their GP to be sufficiently ill that it would make participation difficult. Expressive aphasia was not a basis for exclusion. Twenty-five prospective participants were sent a study invitation pack by each of the five practice administrators, with replies being sent to the study team. The invitation packs included a covering letter, a participant information sheet, a reply slip, and a short questionnaire requesting demographic information, and details of their stroke (how long ago; level of disability). Respondents who indicated they were willing to take part were then selected using maximum variety sampling (at least three patients from each practice; at least two from each tertile of Index of Multiple Deprivation score of the East of England; at least eight of each sex; at least two participants from each age range: <65, 65–79, 80 years or over; at least two participants who had a stroke within the previous year and two who had their stroke over a year previously). Twenty-two participants were interviewed (table 1).

### Data collection
Participants who indicated they were willing to take part who were selected for inclusion were contacted by telephone by a researcher to answer any questions they might have. If the participants were still willing to proceed, then a suitable interview time was arranged. Before the interview started, the researcher confirmed that responses would be anonymous and checked what might be done to make the interview as easy as possible for the participant (eg, presence of a carer; where the interviewer sat). A consent form was signed. Interviews took place from February to May 2016.

Interviews were conducted in participants' homes. Interviews were performed by an experienced qualitative researcher who was not a healthcare professional and was not known to any of the participants. An interview guide was used with questions focused on life after stroke. This included questions on family and social relations, the participants' sense of self, their emotions, interactions with healthcare professionals and how they reflected on their experiences. Questions addressed both the nature of small practical adjustments required and participants' wider sense of their experiences. Follow-up questions were used to probe for relevant details.

## Patient and public involvement

A patient and public involvement (PPI) member was involved in developing the research as a coapplicant. The results of the interviews were fed back to a multidisciplinary intervention development group that included PPI representation who helped place the results in context. The research was overseen by an independent steering committee that included a PPI member. The open-ended questions used in the interviews were informed by meetings with community stroke survivor and carer support groups.

## Thematic analysis

All interviews were recorded and transcribed verbatim by professional transcribers prior to analysis. All transcripts obtained for the original study were included in this secondary analysis. Thematic analysis (TA) is a 'technique for identifying, analysing and reporting patterns (themes) within data'.[25] Researchers using TA have praised its capacity to tease out contradictions between statements saying one thing and statements suggesting another, and to explore how people construct and give account of their attitudes.[26] TA was combined with a constant comparative approach, using existing theory and literature from coping and qualitative work on life after stroke to sensitise our development of codes, theme generation and analysis.

Two coauthors collaborated on the development of the codebook for the TA, which contained the code, a definition, guidelines for when and when not to use the code and examples.[27] The first author then coded all of the verbatim interviews using NVivo software. Themes were identified as aggregates of codes, speaking to broader ideas in the literature. These themes were revised and refined several times by looking at how they held up as valid interpretations of the data when referring back to the transcripts, and on the basis of discussion with the rest of the study team.[28]

## FINDINGS

Participants all spoke about their preventative coping efforts. Some employed the word 'coping' itself while others described how they evaluated and dealt with the challenges of secondary prevention using different terms.

What was most striking about the accounts was the strong emphasis of participants on individual efforts. There was little explicit acknowledgement that coping might be a jointly pursued activity, though this was implicit. Rather, the two most prominent themes with respect to secondary prevention were (a) the need for the self to strive to be 'good', and (b) the role of 'personality' as a resource for these efforts.

### Striving to be 'good'

Across the interviews, participants employed the language of discipline: 'don't eat silly things', 'strict', 'moderation', 'it's me own fault', 'control', 'careful'. This underwrote their depictions of preventative behaviour in moralistic terms, and the weight of expectation they felt to be 'good'. Participants often discussed how doctors had evaluated their efforts to change their health-affecting behaviours and habits such as eating, drinking and exercise. One interviewee relayed the advice given to her about losing weight while coupling it with the doctor's approval of her habits and behaviours:

> They did say to me 'Try and lose a little bit of weight', but you know I mean he came, the consultant came in on the day I came home and he said 'This is where I should be saying to you stop smoking, stop drinking, there's your cholesterol tablets and there's your blood pressure tablets' and he said 'I can't' because I don't smoke, I've never smoked, the occasional drink and the other two were fine so he said 'You know, try and get a little bit of weight off and just have your exercise,' and yeah. (Woman, aged 45–54)

The participant here depicts their successful prevention efforts, constructed in contrast to the hypothetical and less adherent patient that her doctor described. Another patient described interactions with his doctor about changing his drinking behaviour to better prevent a stroke:

> I wasn't happy with the doctor I saw, she kept telling me oh you can have another stroke very quickly, you know. Yeah you'll soon have another stroke if you don't do this and you don't do that. If you do too much you'll have another stroke, you know that don't you. If you do…and this was continual, this woman kept telling me this. I do drink more than I should do, I admit that, not a vast amount, but I do drink more than I should do. I drink wine, and I do enjoy a nice drink. But she said oh, if you have…drink, you know, more than one glass full you're going to have a stroke. (Man, aged 55–64)

The two accounts above, as with most of our interviews, depicted doctors as external adjudicators of individuals' coping efforts. Though one (woman, aged 45–54) speaks of broadly being affirmed and the other (man, aged 55–64) of feeling chastised for a perceived lack of effort, they both draw on doctors as arbitrating the validity of their attempts to prevent a future stroke and how 'good'

they have been. In contrast to the prominent role given to doctors, other healthcare professionals and family members were mentioned in other parts of the interview but not generally in discussions of secondary prevention.

One curious aspect of the accounts was that, despite commonalities in the language of discipline used, survivors did not necessarily give the same description of what being 'good' or 'bad' entailed. Participants were variously concerned with striving to be good in their medication, smoking, weight, exercise or alcohol use—or some combination of the above, though very rarely with more than a few. What the accounts had in common was the conception that participants felt that they should be *striving* for these changes or behaviours—even if they currently fall short or disagree with their doctor's judgement of success. This theme suggests that participants construct and use categories of 'good' and 'bad' behaviours in relation to prevention, and that moralistic terms are even used by those survivors feeling they have behaved in ways that are not 'good'.

### Appeal to personality

A consistent theme throughout the interviews was survivors' appeal to the idea of 'personality' or more generally stable traits and qualities in accounting for their coping efforts. Participants consistently described the self as fixed rather than a malleable set of traits. They perceived this stability as of fundamental importance in hindering or helping their preventative coping attempts, and in explaining how their experiences prior to stroke shaped the way they dealt with the challenges they faced. One of the women in the sample was asked how she had coped emotionally since the stroke:

> I'm not a very emotional person, I don't think, really. I mean, I just think things in my head, but I never cry, never, ever cry. I don't know why, I think that's something, I don't know why, I mean, I used to when I was younger, but nothing seems to, perhaps I've seen too much and done too much, I don't know, I never sit and cry over anything. (Woman, aged 75–84)

She attributed her emotional coping efforts to the fact that she is 'not very emotional', a quality described in irreversible terms.

Other participants identified their professional lives as being particularly formative for their personality, and therefore their coping strategies. Several men spoke of their time in the military, linking the job to their mindset, behaviours and physicality: 'I was in the military police for years and being in the military, it's a life…I was getting on with things, you know you just make do and make the best of the situation' (Man, aged 55–64). This man illustrated 'getting on with things' by describing the sit-ups and press-ups he did.

The role of personality was also emphasised by those describing their struggles to manage after stroke. They spoke of their personal traits and capabilities as stable from before to after their stroke, but now in conflict with the conditions of their life. One survivor said during her interview that being dependent was 'against my nature' and that the occurrences of reliance on others provoked sadness as it reminded her of 'everything that happened'. She spoke about the challenges she faced in maintaining previous roles:

> Just very frustrated, helpless really, I think frustration because I'm not the kind, I've never been the kind of person to ask for help, you know, I'd always just got on and done things, you know and I was having to rely on, you know, my husband to, I'm looking at him and saying, and he's having to put the words there for me and I'm saying 'Yeah, that's what I meant', you know and then I think about a week afterwards I said to my son 'Oh would you like, would you like an omelette for tea?' And he said 'Yeah, but I can do that mum', I said 'No, I'm going to do you an omelette', and I went in the kitchen and he said 'What's the matter?' And I went '[tuts] I don't know how to do it'. (Woman, aged 45–54)

The repetition of the dialogue with her son about making an omelette signifies the salient gap between her perceived personal qualities and her capabilities after stroke, a gap that runs contrary to her 'nature' to be independent, active and contributing to the care of others rather than receiving care herself. Appeal to fixed personal qualities served in this way as a narrative resource for explaining the meaning of present challenges, through reference to life before the stroke. The continuity of personality provided a way of both displaying and appraising discontinuities.

### DISCUSSION

Compared with research on coping with current threats, there has been much less attention to preventative coping. We sought to examine how stroke survivors characterise their coping process in the context of secondary prevention. In our data, themes of personal responsibility and self-sufficiency featured strongly. So did the figure of the doctor as adjudicator of being a 'good' stroke survivor. When compared with the other challenges faced by our participants, secondary prevention appeared particularly 'medicalized': both the criteria for complying and the judgement of success were characterised as external to participants, though the extent of success was held to be a fundamentally individual matter.

In this regard, we were struck that no participant in this study mentioned advice or conversations with healthcare professionals regarding the possible emotional effects of being at increased risk of a future stroke. Doctors loomed large in the imaginations of our participants, but are not sensed to be especially accessible. Other healthcare professionals were mentioned at times, but not characterised as relevant to preventative coping efforts. This finding regarding secondary prevention aligns with previous studies which have identified that stroke survivors living

in the community often report they have little support from healthcare providers for recovery.[4 29] It is interesting to place this together with findings that suggest healthcare professionals perceive themselves as attentive and responsive to the emotional effects on patients of being at increased risk of a future stroke,[30] signalling a potential disconnect between the perceptions of patients and professionals.[31 32]

More generally, our findings align with the conclusions of other qualitative studies, which have highlighted constructions of self-responsibility and the doctor as adjudicator in the narratives of survivors about recovery. Our work extends these findings by showing that the themes continue to influence accounts of efforts to reduce subsequent cardiovascular risk. However, we were also struck by some differences in how these themes resonated in our data compared with accounts, influenced by Foucault, in the existing literature. Disciplinary power was not operating, as for Foucault, to remake the individuals in the image proposed by medical knowledge as normal as opposed to abnormal. Rather, self-responsibility was frequently asserted through emphasis on continuity, rather than on remaking the self. A primary means used by survivors to characterising their efforts to be 'good' was to highlight the strengths they could draw upon from their life before stroke. In his late work, it is notable that Foucault himself found his characterisation of discipline too monolithic, and advocated the need to attend closely to exactly how self-responsibility was being conceptualised. He argued that efforts to strive to be good might operate quite differently, depending on what is construed as the 'ethical substance': 'the way in which the individual has to constitute this or that part of himself as the prime material of his moral conduct'.[33] In our data, the prime material of the survivor was not constituted as requiring change in order to pursue preventative coping: rather, the resources made available by a stable personality were characterised as needing to be turned towards these challenges.

This narrative emphasis on continuity of the self stood in contradiction to the numerous implicit ways that speakers acknowledged major disruptions in their lives and self-concept. This is consistent with previous findings on how survivors talk about life after stroke. Previous studies have documented that stroke survivors' sense of biographical continuity is shaped by the extent to which the stroke has impacted their self-image.[34–36] Our findings suggest a bidirectional relationship, in which self-image may be preserved through an insistence on continuity, and the sequestration of recognition of change.

Participants characterised their personality traits as something they drew on as individuals to help them in preventative coping. The descriptions were of these traits as being external to and unaffected by their stroke, even if this trait cannot be expressed in the same way as before the stroke. Such accounts may reflect cultural understandings, put forward in 20th-century psychology and circulated into popular discourse, in which traits

are essential and independent of what they encounter.[37] Some of those that described struggling in their coping efforts, as in previous research,[3] contrasted their conception of the former efficacy of their own personal traits and abilities with what they struggled with now. The disruption for them appears to be the inability to behave in ways congruent with their personality. It seems plausible that if stroke survivors consider personal traits as constants in their lives, those that then describe their inability to function or participate as they had previously may consider themselves at fault in some way for their own condition. Furthermore, behaviour change interventions or forms of therapy such as acceptance and commitment approaches could seem futile to participants if these changes would conflict with their notion of 'personality'. Our findings regarding the importance of narrative appeal to an essentialised notion of personality contribute to the growing literature exploring potential barriers and facilitators to changes in self-concept among stroke survivors.[38]

### Limitations

There are several limitations to our study relevant to the interpretation of our results. First, our interviews were with individual survivors. This may have foregrounded the theme of individual responsibility in our work, as in other qualitative studies. It may be anticipated that had dyadic interviews been undertaken, whether the survivor+family member or survivor+healthcare professional, the results would have foregrounded joint coping efforts towards secondary prevention.[39] Likewise, in our study and in other qualitative studies, the theme of individual striving to be good with the doctor as adjudicator may also have been less prominent or figured quite differently if dyadic interviews had been undertaken. For instance, the role of family members as additional adjudicators may have been more prominent in a joint interview between survivor and spouse. Our findings therefore relate primarily to the individual's account of their preventative coping efforts, and there may be important differences to the assumptions that guide joint coping activities with others, most notably family members.

This raises the wider limitation that our narrative data are from a single time point and lacking triangulation with observational findings. So, for instance, our reflections on the potential for a static image of personality to hinder utilisation of behaviour change interventions or update their self-concept in adaptive ways therefore remain speculative. There may also be relevant limitations in the transferability of our findings across healthcare settings, including culturally specific aspects of our participants' accounts. While accounts by stroke survivors of the need to strive to be good, with doctors as adjudicators, have been reported across various countries, our study is the first to highlight the theme of personality traits in survivors' accounts of preventative coping. Pickersgill has proposed that psychological discourses of personality have become prominent in the UK.[37] So we do not know the extent to which this finding may have been shaped

by the national context, or whether essentialised notions of self, if not the idea of personality, are relevant more widely to how survivors think about preventative coping.

## CONCLUSIONS

Our study reported from interviews with stroke survivors, conducting a TA to explore preventative coping. Two themes were generated from analysis of the transcripts. A first theme was of the individual's efforts to strive to be 'good', as adjudicated by doctors. Coping was figured as an individual's task, with little reference to joint efforts with family members or healthcare professionals. Indeed, doctors loomed large in the imaginations of our participants, but were not sensed to be especially accessible. Family members and other healthcare professionals, while discussed in other contexts, were not generally characterised as contributing to preventative coping efforts. The theme of individualised efforts to be 'good' aligns with other interview studies of survivors' rehabilitation, extending this observation to the context of secondary prevention. A second, novel theme was of reference by participants to 'personality' as a source of felt continuity and a resource for preventative coping. Appeals to essentialised notions of self may help survivors sustain feelings of biographical coherence, but may also prove an obstacle to behaviour change, as well as adaptive alterations in self-concept.

**Contributors** The study was conceived and designed by PSR, EN, FC, RM, JM and RD. PSR, FC, RM, JM and RD are academic researchers. EN was a student intern. Transcripts were coded by PSR and EN. PSR, RD and JM drafted the manuscript. RD was the guarantor. All authors have read and approved the final manuscript.

**Funding** This study is funded by the National Institute for Health Research's (NIHR) Programme Grant for Applied Research titled 'Developing primary care services for stroke survivors' (reference: PTC-RP-PG-0213–20001).

**Disclaimer** The views expressed are those of the authors and not necessarily those of the NIHR or the Department of Health and Social Care. JM is an NIHR senior investigator.

**Competing interests** None declared.

**Patient and public involvement** Patients and/or the public were involved in the design, or conduct, or reporting, or dissemination plans of this research. Refer to the Methods section for further details.

**Patient consent for publication** Obtained.

**Ethics approval** This study involves human participants. Favourable ethical opinion for the research was gained on 5 November 2015 from East of England Cambridge South Research Ethics Committee (IRAS project ID: 182927REC, reference: 15/EE/0374). Participants gave informed consent to participate in the study before taking part.

**Provenance and peer review** Not commissioned; externally peer reviewed.

**Data availability statement** Data are available upon reasonable request. Please contact JM (jm677@medschl.cam.ac.uk).

**ORCID iD**
Robbie Duschinsky http://orcid.org/0000-0003-2023-5328

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
