## [Reviewer comments · BMJ Open]

ARTICLE DETAILS

TITLE (PROVISIONAL)	Accounts of preventative coping: An interview study of stroke survivors on General Practice registers
AUTHORS	Scott Reid, Peter; Neville, Emma; Cater, Frances; Mullis, Ricky; Mant, Jonathan; Duschinsky, Robbie

VERSION 1 – REVIEW

REVIEWER	Iseult Wilson Queen's University Belfast, Nursing and Midwifery
REVIEW RETURNED	24-Nov-2021

GENERAL COMMENTS	Thank you for this paper, and it was a pleasure to read. You have introduced some interesting concepts in terms of how people who have survived a stroke, might approach prevention of another stroke. I hope you are planning to do more work in this interesting area. Overall, this paper was well-written. My main comments are around some omissions in the methods, and I believe these can be easily addressed. The introduction was clear, flowed well, and included a clear rationale for the study. Methods: There is some detail missing in the methods, and if added, will demonstrate greater rigour. It is possible that the details are in a different paper, as I note that this study is a secondary analysis. Even so, that paper (i.e. the paper presenting the findings from the first study) was not referenced, and I think it would be useful if the details were included here. I suggest looking at the COREQ guidelines, and although some of the items in the checklist have been included, many, especially in the first two domains, were not. ISSM_COREQ_Checklist.pdf (elsevier.com) I have highlighted a few points below. You refer to 3 geographical areas and practices within those areas. You also mention that a register might hold 150 or more cases. In terms of the first study (the actual interviews), it is possible, therefore, that there were hundreds of people who were screened. Clarifying this number and explaining how you reached a total of 21 participants from this number, would help clarify the sampling strategy. There was no mention of how your participants were approached, whether there were any participant information sheets, or how informed consent was obtained. You did mention that you were rigorous in ensuring informed consent, so there is only a little more detail needed here. You also stated that you received ethical permission, however, clarifying the details above would not only help the readership understand the processes you undertook, but also help others who may be interested in conducting similar research. In relation to this study, (i.e. the secondary analysis), it would be useful to add a sentence or two explaining how you gathered your data. For example, did you select all or only some of the transcripts from
---

	the previous study, and your rationale for this. The other information that is missing in relation to the data gathering, is who conducted the interviews, whether they were known to the participants, and if so, how? In qualitative research, because the researcher is so integrally involved in the data collection and analysis, it is very important that the reader knows that researcher/participant relationship. Was the interviewer a GP known to the participants? Was a reflexivity exercise undertaken, and how were biases and assumptions managed? Data analysis and results: The strategy you used was clear and transparent, and your points were well supported by appropriate quotes from a few of the participants in your study. Discussion: This was very interesting and you present some thought-provoking ideas around perceptions, especially around the concept that behaviour changes approaches might be in conflict with a participant's notion of personality. One small editing error: Page 18 line 10/11: lives There were some minor inconsistencies in the formatting of the reference list.
--	---

VERSION 1 – AUTHOR RESPONSE

Reviewer comments	
Thank you for this paper, and it was a pleasure to read. You have introduced some interesting concepts in terms of how people who have survived a stroke, might approach prevention of another stroke. I hope you are planning to do more work in this interesting area. Overall, this paper was well-written. My main comments are around some omissions in the methods, and I believe these can be easily addressed. The introduction was clear, flowed well, and included a clear rationale for the study.	Thank you for this positive feedback.
Methods: There is some detail missing in the methods, and if added, will demonstrate greater rigour. It is possible that the details are in a different paper, as I note that this study is a secondary analysis. Even so, that paper (i.e. the paper presenting the findings from the first study) was not referenced, and I think it would be useful if the details were included here. I suggest looking at the COREQ guidelines, and although some of the items in the checklist have been included, many, especially in the first two domains, were not. ISSM_COREQ_Checklist.pdf (elsevier.com) I have highlighted a few points below.	As requested by the editors, we have reviewed our adherence to the Standardised Reporting of Qualitative Research Checklist (SRQR). Thank you for highlighting the key issues below.
You refer to 3 geographical areas and practices within those areas. You also mention that a register might hold 150 or more cases. In terms of the first study (the	We have expanded our section describing the selection of study participants. In hindsight, the information that we did present was confusing, so we

actual interviews), it is possible, therefore, that there were hundreds of people who were screened. Clarifying this number and explaining how you reached a total of 21 participants from this number, would help clarify the sampling strategy.	have also re-written as well as expanded: The aim of the original study was to understand how primary care meets stroke survivors' needs, and how it could better address them in the future. Participants were recruited from five General Practices in the East of England. Our target was to recruit 20 stroke survivors. The participating General Practices generated a list of people on their stroke register, and a random sample of these were reviewed by the local general practitioner to confirm eligibility. P6 Twenty five prospective participants were sent a study invitation pack by each of the five practice administrators, with replies being sent to the study team. Respondents who indicated they were willing to take part were then selected using maximum variety sampling (at least 3 patients from each practice; at least 2 from each tertile of Index of Multiple Deprivation score of the East of England; at least 8 of each sex; at least 2 participants from each age range: < 65 years; 65-79 years; 80 years or over; at least 2 participants who had a stroke within the previous year and 2 who had their stroke over a year previously. P6
There was no mention of how your participants were approached, whether there were any participant information sheets, or how informed consent was obtained. You did mention that you were rigorous in ensuring informed consent, so there is only a little more detail needed here. You also stated that you received ethical permission, however, clarifying the details above would not only help the readership understand the processes you undertook, but also help others who may be interested in conducting similar research.	We have now provided this detail, see immediately above, and below: The invitation packs included a covering letter, a participant information sheet, a reply slip, and a short questionnaire requesting demographic information, and details of their stroke (how long ago; level of disability). P6 Participants who indicated they were willing to take part who were selected for inclusion were contacted by telephone by a researcher to answer any questions they might have. If the participants were still willing to proceed, then a suitable interview time was arranged. Before the interview started, the researcher confirmed that responses would be anonymous and checked what might be done to make the interview as easy as possible for the participant (e.g. presence of a carer; where the interviewer sat). A consent form was

	signed. P7
In relation to this study, (i.e. the secondary analysis), it would be useful to add a sentence or two explaining how you gathered your data. For example, did you select all or only some of the transcripts from the previous study, and your rationale for this.	We have added the following: All transcripts obtained for the original study were included in this secondary analysis. P8
The other information that is missing in relation to the data gathering, is who conducted the interviews, whether they were known to the participants, and if so, how? In qualitative research, because the researcher is so integrally involved in the data collection and analysis, it is very important that the reader knows that researcher/participant relationship. Was the interviewer a GP known to the participants? Was a reflexivity exercise undertaken, and how were biases and assumptions managed?	The researcher who conducted the interviews was not a health care professional and did not know any of the participants. We have added the following: Interviews were performed by an experienced qualitative researcher who was not a health care professional and was not known to any of the participants. P7
Data analysis and results: The strategy you used was clear and transparent, and your points were well supported by appropriate quotes from a few of the participants in your study.	Thank you.
Discussion: This was very interesting and you present some thought-provoking ideas around perceptions, especially around the concept that behaviour changes approaches might be in conflict with a participant's notion of personality.	Thank you
One small editing error: Page 18 line 10/11: lives	We have corrected 'live' to 'life' on p13.
There were some minor inconsistencies in the formatting of the reference list.	We have added the web-link to the National Clinical Guidelines for Stroke, which was the only inconsistency we could spot.

VERSION 2 – REVIEW

REVIEWER	Iseult Wilson Queen's University Belfast, Nursing and Midwifery
REVIEW RETURNED	25-May-2022
GENERAL COMMENTS	Thank you for addressing my previous comments, and I wish you every success in your work.